# Metformin Enhances Nomegestrol Acetate Suppressing Growth of Endometrial Cancer Cells and May Correlate to Downregulating mTOR Activity In Vitro and In Vivo

**DOI:** 10.3390/ijms20133308

**Published:** 2019-07-05

**Authors:** Can Cao, Jie-yun Zhou, Shu-wu Xie, Xiang-jie Guo, Guo-ting Li, Yi-juan Gong, Wen-jie Yang, Zhao Li, Rui-hua Zhong, Hai-hao Shao, Yan Zhu

**Affiliations:** 1Pharmacy School, Fudan University, Shanghai 200032, China; 2Lab of Reproductive Pharmacology, NHC Key Lab of Reproduction Regulation, Shanghai Institute of Planned Parenthood Research, Fudan University, Shanghai 200032, China

**Keywords:** progestins, nomegestrol acetate, metformin, endometrial cancer, apoptosis, mTOR signaling

## Abstract

This study investigated the effect of a novel progestin and its combination with metformin on the growth of endometrial cancer (EC) cells. Inhibitory effects of four progestins, including nomegestrol acetate (NOMAC), medroxyprogesterone acetate, levonorgestrel, and cyproterone acetate, were evaluated in RL95-2, HEC-1A, and KLE cells using cell counting kit-8 assay. Flow cytometry was performed to detect cell cycle and apoptosis. The activity of Akt (protein kinase B), mTOR (mammalian target of rapamycin) and its downstream substrates 4EBP1 (4E-binding protein 1) and eIF4G (Eukaryotic translation initiation factor 4G) were assayed by Western blotting. Nude mice were used to assess antitumor effects in vivo. NOMAC inhibited the growth of RL95-2 and HEC-1A cells, accompanied by arresting the cell cycle at G0/G1 phase, inducing apoptosis, and markedly down-regulating the level of phosphorylated mTOR/4EBP1/eIF4G in both cell lines (*p* < 0.05). Metformin significantly increased the inhibitory effect of and apoptosis induced by NOMAC and strengthened the depressive effect of NOMAC on activity of mTOR and its downstream substrates, compared to their treatment alone (*p* < 0.05). In xenograft tumor tissues, metformin (100 mg/kg) enhanced the suppressive effect of NOMAC (100 mg/kg) on mTOR signaling and increased the average concentration of NOMAC by nearly 1.6 times compared to NOMAC treatment alone. Taken together, NOMAC suppressing the growth of EC cells likely correlates to down-regulating the activity of the mTOR pathway and metformin could strengthen this effect. Our findings open a new window for the selection of progestins in hormone therapy of EC.

## 1. Introduction

Endometrial cancer (EC) is the most common gynecological malignancy in developed countries and its incidence is rising worldwide due to increased obesity rates [1,2]. In 2019, there will be an estimated 61,880 new cases of and 12,160 deaths from EC [3]. It is the only one of gynecologic cancers with increasing incidence and mortality [4]. Generally, EC can be divided into two main types based on clinical and endocrine features. Type I EC is low-grade, endometrioid, non-*TP53* mutative, and hormone receptor positive expression, which usually has a good prognosis. By contrast, type II EC is characterized by high grade, *TP53* mutative, and hormone receptor negative expression with poor outcome and high lethality [5,6,7]. 

Although the majority of EC patients are diagnosed at an early stage and successfully treated by hysterectomy [8], limited treatment options are available for advanced or recurrent disease and for those who wish to remain fertile. When endometrial cancer is diagnosed in patients of reproductive age, the standard surgical option of hysterectomy and bilateral salpingo-oophorectomy may not be an ideal option [9]. For these patients, hormone therapy could be a better choice since many endometrial cancers are hormonally driven, and hormone therapy relatively lacks toxicity compared to current chemotherapy and radiotherapy [4]. Several derivatives of progesterone have been used for the treatment of advanced and recurrent EC, or patients who wish to preserve fertility [7,10]. Medroxyprogesterone acetate (MPA) and levonorgestrel-releasing intrauterine devices (LNG-IUD) are currently used for hormone therapy of EC in clinic, but the overall response rates of patients with different pathological types and stages towards progestin therapy vary greatly (11–56%) [6]. The response rate to hormone therapy is even lower in advanced (approximately 15–20%) and recurrent patients (nearly ≤10%) [7]. There is a need to search for more effective medicine to treat EC. 

Progestins were previously considered to bind to progesterone receptors (PR) and exert inhibitory effect through down-regulating estrogen receptors (ER) and activating enzymes involved in estrogen metabolism [11]. Recent studies reveal that progestins are able to produce direct and rapid effects on cells and tissues as well via non-genomic mechanisms, and the effects are not suppressed by inhibitors of steroid nuclear receptors [12]. The pI3K–Akt–mTOR (phosphatidylinositol 3-kinase- protein kinase B- mammalian target of rapamycin) pathway belongs to one of these non-genomic mechanisms [12] and has been confirmed to be highly expressed in the tissue of EC [13,14]. Additionally, activated mTOR was reported to promote progestin resistance (usually results from long-term use of progestins). Suppressing the mTOR pathway can inhibit the growth of tumors by inhibiting cell proliferation and promoting cell apoptosis and autophagy [15,16] and reverse progestin resistance in EC cells [17,18]. As a result, an mTOR inhibitor was regarded as a potential target for EC therapy [19,20]. 

Diabetes mellitus has been recently considered as a complication of EC and increase the risk of EC [2]. Metformin, an antidiabetic drug, was found to inhibit the growth of EC cells and sensitize EC cells to chemotherapy at a cellular level [21,22]. Metformin was reported to suppress the activity of mTOR and improve the expression of PR in vitro [23]. Clinically, metformin inhibited EC relapse after MPA therapy [24] and may prolong the overall survival of patients with EC [25,26], but the effects of adjunct metformin were not confirmed in prospective controlled trials [26]. Currently, there is only one experimental paper that describes that metformin (250 mg/kg) strengthened the inhibitory effect of MPA on xenograft tumors of nude mice loaded with Ishikawa EC cells [27]. Accordingly, the efficacy of combining metformin and progestins in EC treatment still needs to be studied. 

Nomegestrol acetate (NOMAC) is a highly selective 19-nor progestogen derivative with the ability to bind to progesterone receptors specifically. Its progesterone activity is higher than that of MPA [28]. The longer half-life and less adverse effects of NOMAC lead to it being successfully used in contraception and in the treatment of many hormone-dependent gynecological disorders, including menstrual disturbances, heavy menstrual bleeding, and premenstrual syndrome [28,29,30]. Preliminarily, we found that NOMAC inhibited the growth of RL95-2 EC cells and the inhibitory effect was stronger than that of MPA [31]. Whether or not NOMAC effectively suppresses the growth of other types of EC cells is yet to be probed. Due to heterogeneous features of tumor cells likely affecting patient response and sensitivity to progestins, it is necessary to distinguish the effects of progestins on different types of EC cells to improve the therapeutic effect of hormone therapy. By using second generation sequencing techniques, we preliminarily found that NOMAC down-regulated a number of genes involved in the Akt–mTOR pathway and metformin could strengthen this effect [data not shown].

In this study, hereby, the inhibitory effect of four progestins, including MPA, LNG, NOMAC, and cyproterone acetate (CPA) on the viability of three types of cells (RL95-2, HEC-1A, and KLE cells) were investigated for probing the sensitivity of the progestins. The antitumor effects of NOMAC and its combination with metformin were evaluated via using EC cells in vitro and nude mice with xenograft tumors in vivo. Moreover, the effects of NOMAC and its combination with metformin on the activity of the Akt–mTOR signaling pathway were investigated as well. 

## 2. Results

### 2.1. NOMAC Inhibited the Growth of Both RL95-2 and HEC-1A Cells

Prior to perform the experiment, we detected the features of three cell lines, including RL95-2, HEC-1A, and KLE. The protein expressions of ER-α and PR were observed in RL95-2 and HEC-1A cells but not in KLE cells, and p53 was detectable in HEC-1A and KLE cells but not in RL95-2 cells, as shown as Figure 1. Then, the inhibitory effects of four progestins, including NOMAC, MPA, LNG, and CPA were measured on the growth of the three types of cell lines. The structures of the progestins tested were shown as Figure 2A.

At the concentration range from 1 to 100 μM, remarkably inhibitory effects were observed after the progestins treatment for 48 h but not at 12 and 24 h. The viability of the EC cells after each progestins treatment for 48 h are shown in Figure 2. NOMAC depressed the growth of three cells in a concentration-dependent trend and the impedance was at its maximum at the concentration of 100 μM (Figure 2B–D). NOMAC exhibited nearly equivalent suppression in RL95-2 and HEC-1A cells with calculated IC_50_ (50% inhibiting concentration) values were between 50~70 μM (Table 1), while weak inhibition was observed in KLE cells with IC_50_ values >200 μM. Comparatively, MPA did not demonstrate a trend of concentration-dependence and its inhibitory rate did not exceed 50% in all EC cells tested (Figure 2B–D). As a result, MPA exhibited weak suppression with its IC_50_ values >200 μM or even no corresponding IC_50_ values could be calculated (Table 1). LNG demonstrated the strongest inhibition in RL95-2 cells with the lowest IC_50_ (95% confidence interval, 95% CI) values of 22.66 (12.17~42.17) μM but extremely weak inhibition in HEC-1A and KLE cells with IC_50_ values >200 μM (Table 1). CPA demonstrated suppression of the growth of RL95-2 cells with IC_50_ (95% CI) values of 40.50 (31.60~51.89) μM but weak inhibition in HEC-1A cells with IC_50_ >100 μM, and no inhibition was found in KLE cells (Table 1).

### 2.2. NOMAC Induced Cell Cycle Arresting and Apoptosis and Down-Regulated the Activity of mTOR–4EBP1/eIF4G in RL95-2 and HEC-1A Cells

Since NOMAC exhibited stronger inhibition in HEC-1A cells than the other three progestins, it was chosen for further investigation.

NOMAC did not alter the distribution of the cell cycle and the percentage of apoptotic cells at the concentration of 10 μM in both RL95-2 and HEC-1A cells. In RL95-2 cells, the DNA contents in G0/G1 and S phase and the percentage of apoptotic cell and dead cells were significantly increased after 30 and 100 μM NOMAC treatments (*p* < 0.05, Figure 3A,B). Especially, the DNA content of G0/G1 and S phase were increased to 28.06 ± 1.41% and 25.72 ± 2.11%, respectively in 100 μM NOMAC treated cells, and the percentage of apoptotic and dead cells were increased to 17.83 ± 1.94% and 39.03 ± 3.55%, respectively, and there was a significant difference compared with the control cells (*p* < 0.05, Figure 3A,B).

In HEC-1A cells, however, NOMAC markedly changed the cell cycle distribution and induced apoptosis merely at the concentration of 100 μM but not at 30 μM. After 100 μM NOMAC treatment, the DNA proportion of G0/G1 and S phase were 47.59 ± 1.81% and 27.01 ± 1.28%, and the percentage of apoptotic and dead cells were up to 22.68 ± 1.28% and 11.57 ± 1.27%, respectively and a pronounced difference was detected compared to the control cells (*p* < 0.05, Figure 3 C,D).

Since the Akt–mTOR (protein kinase B-mammalian target of rapamycin) signaling pathway was preliminarily found to be involved in NOMAC treated EC cells, the effects of NOMAC on activity of several genes in the Akt–mTOR pathway were further investigated. In both RL95-2 and HEC-1A cells, NOMAC did not influence the expression of phospho-Akt (Ser 473) but inhibited the expression of phospho-mTOR (Ser 2448) and its downstream substrate phospho- 4EBP1 (4E-binding protein 1) (Ser 65) and phospho-eIF4G (eukaryotic translation initiation factor 4G) (Ser 1108) in a concentration-dependent trend (Figure 3E,F,G). With increasing concentrations of NOMAC, its suppressive effect on the level of phosphorylated mTOR was enhanced. When the cells were treated with 100 μM NOMAC, the protein level of phospho-mTOR (Ser 2448), phospho-4EBP1 (Ser 65) and phospho-eIF4G (Ser 1108) were significantly down-regulated in both cells. (*p* < 0.01, Figure 3F,G).

For elaborating the influence of NOMAC on mTOR signaling, we further investigated the effect of NOMAC on rapamycin and MHY1485, which are the specific antagonist and the agonist of mTOR, respectively. Rapamycin significantly reduced the activity of mTOR while MHY1485 increased the expression of p-mTOR (*p* < 0.05, Figure 3H,I). Compared with samples treated with rapamycin alone, the inhibitory effects on the level of phospho-mTOR (Ser 2448) and its downstream substrates were markedly strengthened after the cells were simultaneously treated with NOMAC and rapamycin. The promoting effect of MHY1485 on the expression of phospho-mTOR (Ser 2448) and its downstream proteins were impaired under NOMAC and MHY1485 treatment together (*p* < 0.05, Figure 3H,I).

### 2.3. Metformin Enhances the Suppressive Effect of NOMAC and Strengthens Down-Regulation of mTOR–4EBP1/eIF4G in RL95-2 and HEC-1A Cells In Vitro

Since metformin was preliminarily found to strengthen the suppressive action of NOMAC in mTOR pathway, 1 mM metformin was further used to combine with 30 μM NOMAC to investigate how metformin would influence the effect of NOMAC in EC cells.

When 30 μM NOMAC combined with the 1 mM metformin treatment, the cell inhibitory rates were promoted by 12.8% and 17.1% in RL95-2 and HEC-1A cells, respectively (*p* < 0.05, Figure 4A,B), accompanied by a noticeable increase in the percentage of late-stage apoptotic and dead cells in both cells. In RL95-2 cells, the percentage of apoptotic cells and dead cells were increased to 14.92 ± 1.10% and 14.53 ± 2.02%, respectively, and in HEC-1A cells, the percentage of apoptotic cells and dead cells were increased to 8.30 ± 1.38% and 13.87 ± 2.76%, respectively. There were pronounced difference in the cells of control, united treatment, and NOMAC or metformin treatment alone. (*p* < 0.05, Figure 4C,D)

Moreover, combining NOMAC with metformin treatment significantly inhibited the protein levels of phospho-mTOR (Ser 2448), phospho-4EBP1(Ser 65), and phospho-eIF4G (Ser 1108) compared to NOMAC or metformin treatment alone in both RL95-2 and HEC-1A cells (*p* < 0.01, Figure 4E,F,G). However, the expression of phospho-Akt (Ser 473) was not affected even when combining NOMAC with metformin treatments compared to that of control (*p* > 0.05, Figure 4E,F,G).

### 2.4. Metformin Increased the Inhibitory Effect of NOMAC and Strengthened Down-Regulation of Phospho-mTOR/Phospho-4EBP1/Phospho-eIF4G in Mice Model of Xenografts

Since combining NOMAC with metformin treatment significantly inhibited the growth of EC cells in vitro, we further evaluated the antitumor activities of combining NOMAC with metformin in vivo by taking xenografts of nude mice loaded with RL95-2 cells for example.

Two weeks after 28 nude mice were inoculated with RL95-2 cells, solid xenografts were observed in all mice and their volumes were measuring up to 100 mm^3^. Prior to treatment, there were no remarkable differences in tumor sizes among all groups (*p >* 0.05, Figure 5A). After treatment for 24 day, the volume of xenograft tumors in the united treatment group (both NOMAC and metformin were 100 mg/kg, respectively) markedly decreased compared to the control. At the end of 28 d treatment, the volume of xenografts in both the NOMAC 100 mg/kg treatment alone (664.86 ± 470.94 mm^3^) and united treatment group (607.71 ± 438.18 mm^3^) markedly declined compared to the control group (1290.06 ± 341.93 mm^3^) (*p* < 0.05, Figure 5A,B).

No death occurred in mice during treatment, no abnormal behavior, and physiological signs were observed, and no tumor metastasis was found. The body weight of all mice fluctuated in a normal range. Comparing with the control, no obvious differences of body weight and relative liver and kidney weight were observed in treatment groups (*p* > 0.05, Figure 5C,D).

Moreover, the average concentration of NOMAC was 48.46 ng/g in the NOMAC treatment group and 76.29 ng/g in the group of NOMAC and metformin united treatment (Figure 5E).

The result of western blotting showed that combining NOMAC and metformin treatments significantly inhibited the expression of phospho-mTOR (Ser 2448), phospho-4EBP1(Ser 65), and phospho-eIF4G (Ser 1108) compared to NOMAC or metformin treatments alone in the xenograft tumor tissues (*p* < 0.05 or *p* < 0.01, Figure 5F).

## 3. Discussion

In the present study, we found that NOMAC moderately inhibited the growth of both RL95-2 and HEC-1A cells, but MPA, LNG, and CPA did not influence the growth of HEC-1A cells. In RL95-2 and HEC-1A cells, NOMAC arrested the cell cycle at G0/G1 phase, induced apoptosis, and markedly down-regulated the activity of mTOR/4EBP1/eIF4G, and metformin significantly enhanced the inhibitory effect of NOMAC at molecular and cellular levels and increased the average concentration of NOMAC by nearly 1.6 times in xenograft tissues compared to NOMAC treatment alone.

First of all, we confirmed the features of three cell lines, which were consistent with those previously reported [32,33,34,35]. In hormone responsive cells, RL95-2 were usually defined as type I cells while HEC-1A were regarded as type II cells since the former does not strongly exhibit TP53 mutation but the latter does (available online: https://portals.broadinstitute.org/ccle). KLE belong to type II EC cells characterized by hormone receptor negative expression [5] and poor differentiation [35,36]. In the present study, we found that the four progestins generally exhibited strong inhibition in type I but demonstrated disparities in type II cells. Based on the viability of the cells and the value of IC_50_, which usually reflect the inhibitory potency of the medicine tested, NOMAC not only impedes the growth of type I cells but exerts inhibition on HEC-1A cells as well, whereas MPA demonstrated weaker suppression than the other progestins especially in type II cells. MPA was generally used against EC in clinical trials, but its directly suppressive effects on EC cells were seldom mentioned. In the present study, MPA did not demonstrate strong inhibition on the growth of the three types of cell lines, which is consistent with our previous findings [31]. As for LNG and CPA, both of them exhibited stronger inhibitory capability than NOMAC in type I cells but demonstrated weaker inhibition than NOMAC in type II cells. Accordingly, the results suggest that NOMAC targets not only type I cells but also a certain type II cells. NOMAC may exhibit more advantages than the other progestins in the treatment of EC. Here, we compared the inhibitory effects of various progestins on the growth of different type of EC cells. The results may be helpful for clinicians to choose suitable progestins in the treatment of endometrial cancer.

Based on the findings above, the suppressive effects of NOMAC on type I and type II cells were further investigated, taking RL95-2 and HEC-1A cells as examples. It was previously found that NOMAC impeded the proliferation of RL95-2 cells through interfering EdU (5-ethynyl-20-deoxyuridine) -incorporation [31]. Here, we further observed that NOMAC induced cell arrest at G0/G1 phase, promoted apoptosis, and increased the percentage of dead cells in a manner of concentration dependence in both RL95-2 and HEC-1A cells. Taken together, these results suggest that NOMAC not only depressed the DNA synthesis but also induced cell apoptosis in EC cells. The difference between the two cells is that a higher concentration of NOMAC (100 μM) is needed to induce apoptosis in HEC-1A cells, while 30 μM NOMAC was enough to significantly trigger apoptosis in RL95-2 cells. It means that type I cells are more sensitive to NOMAC than type II cells. Apoptosis is regulated by many genes, including the tumor suppressor gene *TP53*. *TP53* can trigger cell death by apoptosis through transactivation of downstream target genes [37,38], but mutations in *TP53* would promote malignant progression and lead to insensitivity of tumor cells toward chemotherapy [39]. Since strong mutation of *TP53* was observed in HEC-1A cells, we presume that the disparities in the two types of hormone responsive cells may partly correlate to the status of the *TP53* gene, but more studies should be done in the future.

Previously, we reported that NOMAC impeded the growth of ectopic endometrium and was associated with depressing the level of hormone receptors [40]. In this present study, NOMAC was further found to significantly depress the phosphorylated protein expression of mTOR and its substrates 4EBP1 and eIF4G in both RL95-2 and HEC-1A cells. mTOR signaling regulates cell functions, including cell growth, apoptosis, and the cell cycle by acting on downstream targets [41,42]. 4EBP1 specifically inhibits cap-dependent protein translation by binding to eIF4E, a eukaryotic transcription initiation factor. Phosphorylation of 4EBP1 induced by mTOR disassociates it from eIF4E, thereby reducing its cap-binding activity and promoting the synthesis of proteins [43]. Since NOMAC strongly depressed the level of phosphorylated mTOR/4EBP1/eIF4G, it is plausible to presume that the inhibitory effect of NOMAC on the cell cycle and DNA synthesis likely correlates to the decrease in the activity of mTOR and its related signaling pathway.

We further explored the role of mTOR in NOMAC treated EC cells. Except for siRNA techniques, the mTOR inhibitor and agonist can be used to preliminarily identify whether or not mTOR is the action site of a medicine [44,45], and therefore, rapamycin and MHY1485, the specific antagonist and agonist of mTOR were used in the experiment. The results demonstrated that NOMAC enhanced the action of the antagonist on mTOR and damaged the effect of the agonist on mTOR. It suggests that the activity regulation of mTOR and its downstream pathway were likely involved in the mechanism of action of NOMAC in both EC cells.

Moreover, the effects of NOMAC combined with metformin were investigated in vitro. According to Mitsuhashi et al. [46] and Takahashi et al. [47], lower concentrations (<1 mM) of metformin did not produce inhibition in Ishikawa and HEC-1B cells. In the present study, we also found that the concentration of metformin was positively correlated to its efficacy. One millimolar metformin produced mildly inhibitory effects in RL95-2 and HEC-1A cells, and therefore, 1 mM metformin was chosen to be used to combine with NOMAC. Additionally, because combining 10 μM NOMAC with 1 mM metformin did not exhibit an obvious increase of apoptosis compared to their treatments alone, 30 μM NOMAC was used to combine with 1 mM metformin in the experiment. The results demonstrated that 1 mM metformin enhanced the suppressive action of 30 μM NOMAC in impeding the growth of cells, inducing cell apoptosis and decreasing the activity of mTOR and its downstream substrates in both cells. This suggests that metformin would strengthen the action of NOMAC on down-regulating the activity of the mTOR signaling pathway. Our results provide an experimental basis for combining metformin with progestins in clinical treatment of EC.

Noticeably, NOMAC did not change the level of phospho-Akt (Ser 473) even if it was combined with metformin in the experiment. This suggests that the decrease of mTOR activity induced by NOMAC might not be related to phospho-Akt (Ser 473) in the tested cells. The activity of mTOR is probably regulated by many other upstream signaling molecules, such as AMPK and ERK [48,49]. The upstream target of NOMAC-induced decrease in mTOR activity needs more experiments to be validated.

In view of the prominently suppressive effects in vitro, the suppressive effects of combining NOMAC with metformin were further evaluated in nude mice bearing RL95-2. The dose of 100 mg/kg metformin and 100 mg/kg NOMAC were combined in the animal experiment, since 100 mg/kg metformin for mice is nearly equivalent to 700 mg for a 60 kg human, which is within the dosage range for human diabetes therapy. No obvious behavioral or physiological changes were observed in mice during the combined treatment. This suggests that the combined treatment is plausible. In the xenograft tissues, the pronounced reduction of the activity of mTOR and its downstream signals 4EBP1 and eIF4G were verified after combining NOMAC and metformin compared to NOMAC or metformin treatments alone, which were consistence with that observed in vitro. It indicates that mTOR and its downstream signaling likely participate the suppressive action of NOMAC.

Clinically, progestins are usually given for a long time (usually longer than six months) in EC hormone therapy [50,51]. In the experiment, we found that the growth rate of the xenograft tumor remarkably slowed down after merely NOMAC treatment alone for 28 day. This indicates that long-term treatment with NOMAC is needed. However, the shrinking of the xenograft tumor was obviously observed after combining NOMAC with metformin treatment for 24 day. This suggests that metformin could enhance the inhibitory action of NOMAC in the animal experiment. Accordingly, the concentrations of NOMAC were measured by LC–MS, and the results revealed that concentration of NOMAC in tumor tissues was increased by nearly 1.6 times in the combined group compared to NOMAC treatment alone. This suggests that metformin is able to increase the concentration of NOMAC in tumor tissues. We presume that the boosting effect of metformin with NOMAC may be attributed to interactions of metabolic enzymes, but this needs to be validated by more experiments in future. Our results could serve as a reference point for dosage and time of administration in clinical trials.

### Conclusions and Limitations

In summary, NOMAC suppressed not only type I but also certain type II EC cells, such as HEC-1A cells. Comparatively, MPA, LNG, and CPA were less sensitive to type II cells than NOMAC. The mechanism of action of NOMAC may involve down-regulating the activity of the mTOR pathway. Metformin exhibits the ability to increase the effect and the concentration of NOMAC. Our findings provide a new avenue for the selection of progestins in hormone therapy of EC. However, there are limitations in the experiment. In future, we will use siRNA techniques to elaborate the role of mTOR and its upstream signaling pathway to uncover the underlying mechanism of action of NOMAC. Since HEC-1A cells were markedly inhibited by NOMAC and metformin in vitro, its efficacy would be tested in animal experiments. Finally, the efficacy and the safety of NOMAC for antitumor use need to be verified in a clinical setting.

## 4. Materials and Methods

### 4.1. Compounds

NOMAC was obtained from Lijiang Yinghua Bio-Pharmaceutical Co. Ltd. (Kuming, China). MPA was obtained from Xianju Pharma (Ningbo, China). Levonorgestrel (LNG) were obtained from Beijing Zizhu Pharmaceutical Co. Ltd. (Beijing, China) and cyproterone (CPA) was purchased from Medchemexpress Co. Ltd. (Monmouth Junction, NJ, USA). Rapamycin and MHY1485 were purchased from Selleck Chemicals LLC. (Houston, TX, USA). Metformin (Met) was purchased from Sigma–Aldrich Co.LLC. (St. Louis, MD, USA).

### 4.2. Cell Culture

The human EC cell lines RL95-2 and HEC-1A were purchased from the American Type Culture Collection (ATCC, Manassas, VA, USA) and were cultured in Dulbecco modified Eagle medium/F12 (Thermo Scientific, Rockford, IL, USA) and McCoy 5A medium (Thermo Scientific), respectively with 10% fetal bovine serum (FBS) (Thermo Scientific). The KLE cell line was obtained from the China Center for Type Culture Collection (Wuhan, China) and was cultured in Dulbecco modified Eagle medium/F12 with 10% FBS. All cells were incubated at 37 °C in a humidified incubator with 5% CO_2_.

### 4.3. CCK-8 Assay for Cell Viability Analysis

The RL95-2, HEC-1A, and KLE cells were seeded in 96-well plates (1×10^4^ cells/well), and treated with NOMAC, MPA, LNG, or CPA at concentrations of 1, 3, 10, 30, and 100 μM for 12, 24, and 48 h or Met (1 mM), NOMAC (30 μM), and NOMAC (30 μM) plus Met (1 mM) for 48 h. The control cells were treated with the same volume of dimethyl sulphoxide (DMSO) (Sigma–Aldrich, St. Louis, MO, USA). Subsequently, cell viability was detected by cell-counting kit-8 (CCK-8) (Dojindo Molecular Technologies, Kumamoto, Japan) according to the manufacturer’s protocol. The absorbance (optical density, OD) was read with a spectrophotometric plate reader (BioTek ELX-800, Winooski, VT, USA) at 450 nm. Cell viability (%) was calculated using the following equation: cell viability (%) = OD of treatment cells/OD of control cells × 100%. Inhibition rate (%) = 1 – OD of treatment cells/OD of control cells × 100% (%). Final results were presented as half maximal inhibitory concentration (IC_50_) with 95% confidence intervals (95% CI), which were calculated from the nonlinear regression model based on log(inhibitor) vs. normalized response/variable slope dose response curves using GraphPad Prism 6.02 (GraphPad Software Inc., San Diego, CA, USA).

### 4.4. Animal Studies

Twenty eight female athymic nude mice of 18 g were purchased from Shanghai Slike Experimental Animal Co. Ltd. (Shanghai, China) and housed under specific-pathogen-free conditions at 25 °C with 60 ± 10% humidity. After acclimating to the environment for 1 week, mice were subcutaneously injected with RL95-2 cells (1 × 10^7^ cells/mouse) on the right flank. When tumors reached a volume of about 100 mm³, mice were randomly divided into 4 groups (7 mice/group) according to the volume of xenografts and received the following interventions via intragastric gavage once daily for 4 weeks: solvent control, 100 mg/kg metformin, 100 mg/kg NOMAC, and 100 mg/kg metformin + 100 mg/kg NOMAC. NOMAC was dissolved in normal saline solution with 0.5% sodium carboxymethylcellulose (CMC-Na) (Sinopharm Chemical Reagent Co. Ltd., Shanghai, China). Tumor volume was measured twice a week using a digital caliper and was calculated with following formula: V(mm^3^) = 1/2 (L × W × W) (V, volume; L, length; W, width). Mice were sacrificed by anesthesia 1 h after the last treatment. Harvested tumors were processed for further analyses. All surgical and experimental procedures were reviewed and approved by the Laboratory Animal Ethics committee at the Shanghai Institute of Planned Parenthood Research (approval no. 2017-07, 16 August 2017).

### 4.5. LC–MS Assay

Tumor tissue samples were weighed and normal saline (0.2 mL/100 mg) was added to prepare tissue homogenate by tissue homogenizer. Then 10 μL internal standard working fluid (levonorgestrel, 50.0 ng/mL) (National Institutes for Food and Drug Control, Beijing, China) was added to 100 μL homogenate and the mixture was vortexed for 1 min (the final concentration of levonorgestrel was 5 ng/mL). The organic phase was separated by 1 mL cyclohexane (General-Reagent, Shanghai, China) and centrifuged at 10000 rpm (10 °C) for 10 min. In a clean centrifugal tube, 0.98 mL of the organic phase was dried with nitrogen at 45 °C. The residue was dissolved with 100 mL initial mobile phase (80% methanol) (MERCK, Kenilworth, NJ, USA) and analyzed by liquid chromatography–mass spectrometry (LC–MS) (Shimadzu, Kyoto, Japan) after 10 min of high-speed centrifugation at 12000 rpm (4 °C).

### 4.6. Annexin V/PI Apoptosis Assay

The RL95-2 and HEC-1A cells were seeded in 6-well flat-bottom microplates at the density of 3×10^5^ cells/well, and treated with NOMAC (0, 10, 30, and 100 μM) or metformin (1 mM), NOMAC (30 μM), or NOMAC (30 μM) plus metformin (1 mM) for 48 h. Then cells were digested by 0.25% trypsin with EDTA (Ethylene Diamine Tetraacetic Acid) and centrifuged at 800 rpm for 5 min. After resuspension with 100 μL binding buffer and labeling by Annexin V and Propidium iodide (BD Biosciences, San Jose, CA, USA), a BD LSRFortessa flow cytometer (BD Biosciences) was used to detect cell apoptosis and the percentage of apoptotic cells was analyzed using Flowjo software (Tree Star, Ashland, OR, USA).

### 4.7. Cell Cycle Analysis

The RL95-2 and HEC-1A cells were seeded in 6-well flat-bottom microplates at the density of 3×10^5^ cells/well, and treated with NOMAC (0, 10, 30, and 100 μM) for 48 h. Then cells were digested by 0.25% trypsin with EDTA and centrifuged at 800 rpm for 5 min. Cells were then washed with PBS twice and fixed overnight in 75% ice-cold ethanol. Fixed cells were then treated with 0.5 mL PI/RNase staining buffer (BD) for each 10^6^ cells and incubated for 15 min in the dark. Flow cytometry was performed to detect the cell cycle distribution.

### 4.8. Protein Extraction and Western Blotting

Cells collected after drug treatment were suspended in high efficiency cell tissue rapid lysis buffer (RIPA) (Invitrogen, Camarillo, CA, USA) containing 1% proteinase and phosphatase inhibitors (Thermo Scientific). Cell lysates were boiled at 100 °C for 5 min and then stored at –20 °C. Protein concentrations were quantified using the BCA protein assay kit (Sangon Biotech, Shanghai, China), total proteins (20 μg for cell samples and 60 μg for tumor tissue samples) were electrophoresed in SDS–PAGE gels and transferred into a PVDF membrane (Millipore, Bedford, MA, USA) for 1.5–2 h. The membranes were blocked with 5% milk in TBST for 1 h and overnight incubated in primary antibody against ER-α (#8644, 66 kDa), PR A/B (#8757, PR-A 90 kDa, PR-B 118 kDa), p53 (#2527, 53 kDa), phospho-Akt (Ser 473) (#9271, 60 kDa), Akt (Pan) (#4691, 60 kDa), phospho-mTOR (Ser2448) (#5536, 289 kDa), mTOR (#2983, 289 kDa), phospho-4EBP1 (Ser65) (#9451, 15–20 kDa), 4EBP1 (#9644, 15–20 kDa), phospho-eIF4G (Ser1108) (#2441, 220 kDa), and eIF4G (#2469, 220 kDa) (1:1000) at 4 °C. Then the PVDF membrane was washed three times with TBST solution, and incubated at room temperature for 1 h in peroxidase-conjugated goat anti-rabbit IgG (Immunoglobulin G) (#7074) (1:3000). After being washed three times with TBST, the protein bands were visualized using the ECL SuperSignal West Femto Detection Kit (Thermo Scientific). All antibodies were purchased from Cell Signaling Technology (Beverly, MA, USA).

### 4.9. Statistics

In this paper we define all of the error bars as standard error of mean (SEM). Data are presented as the means ± SEM. The data were analyzed with GraphPad Prism version 6.02 (GraphPad Software Inc., San Diego, CA, USA). Statistical significance was determined using one-way analysis of variance and Tukey’s multiple comparisons test was used as the post hoc test when the differences among multiple groups were compared. Differences with *p*-values of less than 0.05 were considered statistically significant.

## Figures and Tables

**Figure 1 ijms-20-03308-f001:**
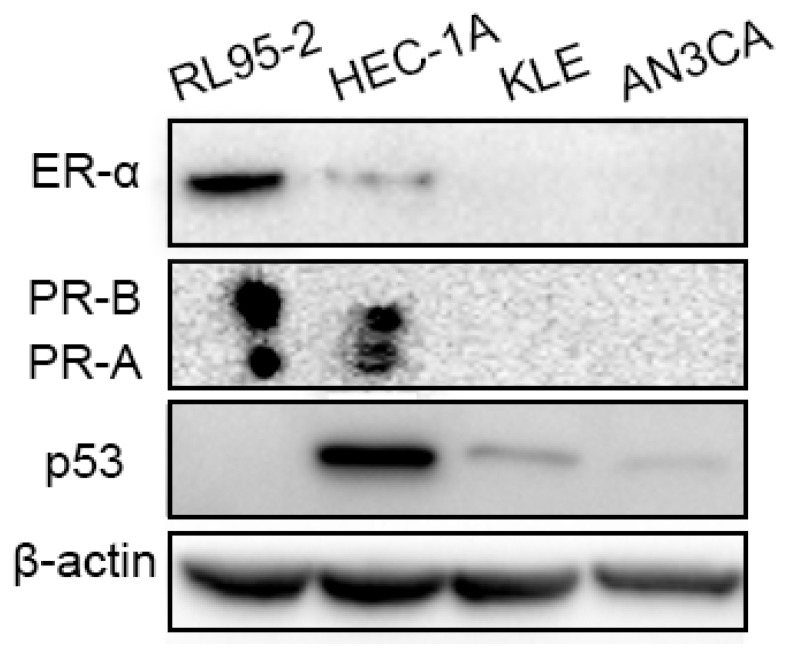
Expression of estrogen receptor-α (ER-α), progesterone receptor (PR), and p53 in RL95-2, HEC-1A, and KLE cells.

**Figure 2 ijms-20-03308-f002:**
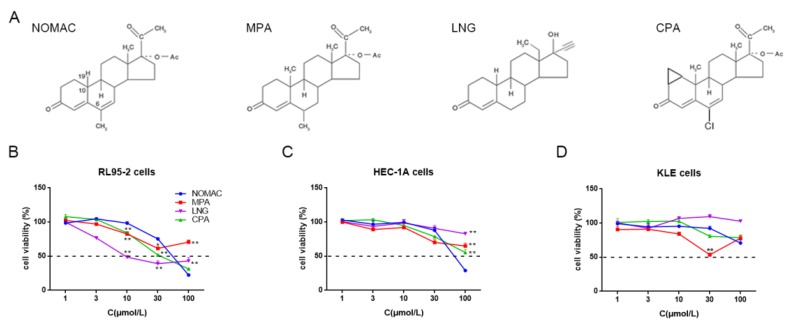
Nomegestrol acetate (NOMAC) simultaneously inhibited the growth of both RL95-2 and HEC-1A cells. (**A**) The structures of nomegestrol acetate (NOMAC), medroxyprogesterone acetate (MPA), levonorgestrel (LNG), or cyproterone acetate (CPA). (**B**, **C** and **D**) Inhibitory effect of four progestins on the viability of RL95-2, HEC-1A, and KLE cells. Cells were treated with NOMAC, MPA, LNG, or CPA at the concentration of 1, 3, 10, 30, and 100 μM for 48 h, respectively. Experiments were performed in triplicate each time and independently repeated three times with different passages of cells. The data were expressed as mean ± SEM. * *p* < 0.05, ** *p* < 0.01 vs. NOMAC treated cells.

**Figure 3 ijms-20-03308-f003:**
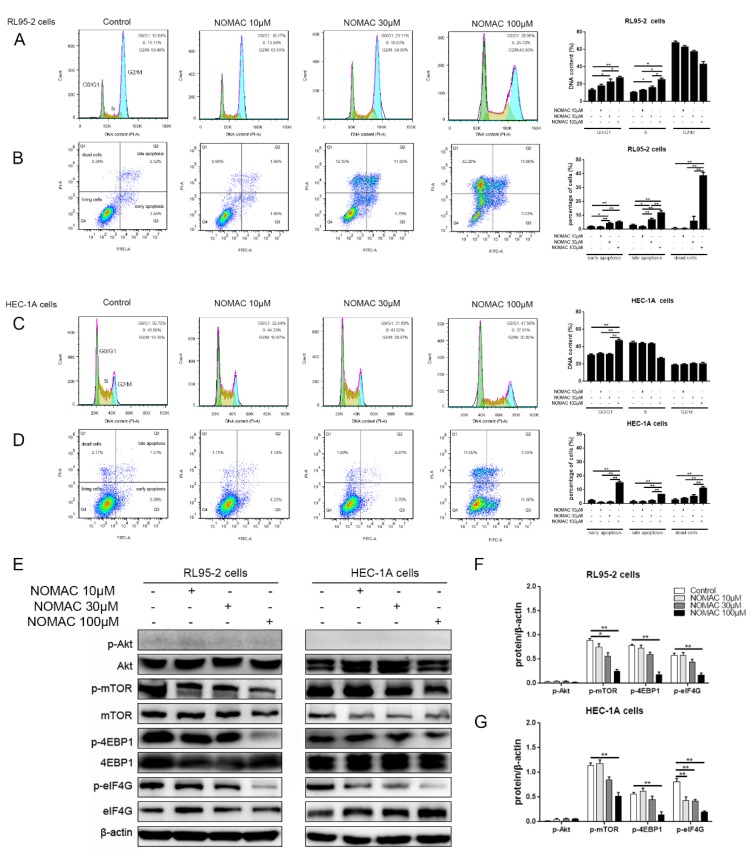
NOMAC induced cell cycle arrest and apoptosis and down-regulated the activity of mTOR–4EBP1/eIF4G (mammalian target of rapamycin-4E-binding protein 1/ eukaryotic translation initiation factor 4G) in RL95-2 and HEC-1A cells. (**A, B, C**, and **D**) The effects of NOMAC on cell cycle and apoptosis in RL95-2 and HEC-1A cells. Cells were exposed to NOMAC (0, 10, 30, and 100 μM) for 48 h. (**E, F**, and **G**) The expression of phospho-Akt (protein kinase B) (Ser 473), Akt (Pan), phospho-mTOR (Ser 2448), mTOR, phospho-4EBP1 (Ser 65), 4EBP1, phospho-eIF4G (Ser 1108), and eIF4G under NOMAC treatment (0, 10, 30, and 100 μM) after 48 h in RL95-2 and HEC-1A cells. (**H** and **I**) The protein expression of phospho-mTOR (Ser 2448), mTOR, phospho-4EBP1 (Ser 65), 4EBP1, phospho-eIF4G (Ser 1108), and eIF4G under NOMAC and rapamycin (mTOR inhibitor, rapa) or MHY1485 (mTOR agonist, MHY) treatments in RL95-2 and HEC-1A cells. Cells were treated with DMSO (as control), rapa 1 nM with or without NOMAC 30 μM, and MHY 1 μM with or without NOMAC 30 μM for 48 h. These results were typical of three independent experiments. The data were expressed as mean ± SEM. * *p* < 0.05 and ** *p* < 0.01.

**Figure 4 ijms-20-03308-f004:**
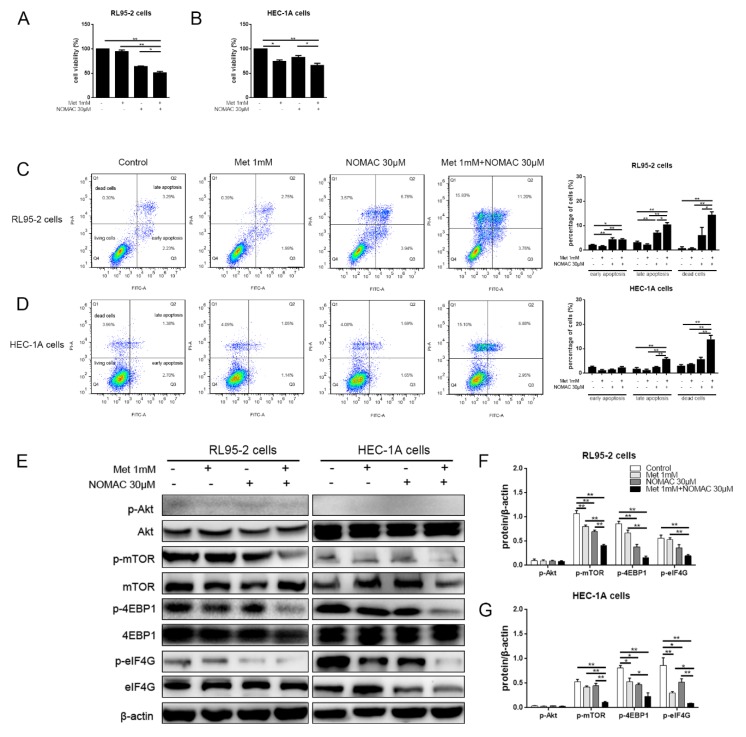
Metformin enhances the suppressive effect of NOMAC and promotes down-regulating of mTOR–4EBP1/eIF4G in RL95-2 and HEC-1A cells in vitro. Cells were treated with NOMAC, metformin (Met), or the combination for 48 h. (**A** and **B**) The CCK-8 assay was performed to analyze cell viability. (**C** and **D**) Cell apoptosis was analyzed using flow cytometry. (**E, F,** and **G**) The expression level of phospho-Akt (Ser 473), Akt (Pan), phospho-mTOR (Ser 2448), mTOR, phospho-4EBP1 (Ser 65), 4EBP1, phospho-eIF4G (Ser 1108), and eIF4G were assessed by Western blot. These results were typical of three independent experiments. The data were expressed as mean ± SEM. * *p* < 0.05, ** *p* < 0.01.

**Figure 5 ijms-20-03308-f005:**
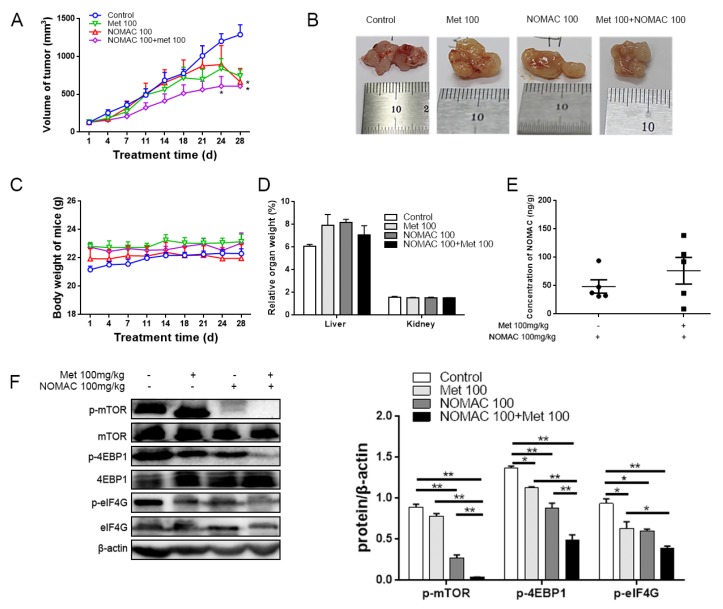
Metformin increased the inhibitory effect of NOMAC and strengthened down-regulation of mTOR–4EBP1/eIF4G in nude mice loaded with RL95-2 cells. Mice were treated with solvent (normal saline solution), 100 mg/kg metformin (Met), 100 mg/kg NOMAC, and 100 mg/kg Met + 100 mg/kg NOMAC (7 mice/group). (**A**) Tumor volume of mice was measured twice a week. * *p* < 0.05 vs. control group. (**B**) RL95-2 xenograft tumor tissues at the end of the experiment. (**C**) Body weight of mice was measured twice a week. (**D**) Liver and kidney were collected and the relative organ weights were compared (relative organ weight = weight of liver or kidney/weight of the mice × 100%). (**E**) Concentration of NOMAC in xenograft tumors. (**F**) Effects of NOMAC used alone or combined with metformin on the expression of mTOR signaling in RL95-2 xenograft tumors. Tumor tissues were collected for analyzing the expression of phospho-mTOR (Ser 2448), mTOR, phospho-4EBP1 (Ser 65), 4EBP1, phospho-eIF4G (Ser 1108), and eIF4G. The data were obtained from three tumor samples with the strongest inhibitory effect in each administration group. All data were expressed as mean ± SEM. * *p* < 0.05, ** *p* < 0.01.

**Table 1 ijms-20-03308-t001:** Antiproliferative activity of various progestins on three types of human endometrial cell lines after treatment for 48 h, as presented as IC_50_ (95% confidence interval) values (μM).

Drug	RL95-2	HEC-1A	KLE
NOMAC	53.47 (48.54~58.90)	65.61 (57.90~74.35)	236.00 (122.90~453.10)
MPA	271.80 (68.10~1085.00)	247.10 (95.33~640.70)	/
LNG	22.66 (12.17~42.17)	/	/
CPA	40.50 (31.60~51.89)	116.40 (87.32~155.30)	/

NOMAC stands for nomegestrol acetate, MPA for medroxyprogesterone acetate, LNG for levonorgestrel and CPA for cyproterone acetate. / means that the corresponding IC_50_ (50% inhibiting concentration) value is higher than 400 μM or could not be calculated out because the viability of cells did not decrease to 50% of the maximum. The data shown are from triple wells in three independent assays.

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
