# Peer review of "Metformin Enhances Nomegestrol Acetate Suppressing Growth of Endometrial Cancer Cells and May Correlate to Downregulating mTOR Activity In Vitro and In Vivo"

_ijms, 2019, doi:10.3390/ijms20133308_

Round 1

Reviewer 1 Report

Congratulations to the authors for providing substantial preclinical evidence to support the beneficial effect of metformin on endometrial cancer which may well serve as a basis for phase II clinical studies.

No change is suggested in the manuscript.

Author Response

Comments and Suggestions for Authors

Congratulations to the authors for providing substantial preclinical evidence to support the beneficial effect of metformin on endometrial cancer which may well serve as a basis for phase II clinical studies.

No change is suggested in the manuscript.

Response: Thank you very much for your comments.

Reviewer 2 Report

very well written and designed study

line 53, 'Meanwhile, progestins are able' not 'is able'

For the statistics you likely did an ANOVA to detect a difference exists in the group but then did a t-test between each group to detect what that difference is, correct?  Just state that you also did the t-test.  If you are comparing test results to each other use a Tukey correction but if comparing to zero you may wish to use a Dunnetts correction.

Author Response

Comments and Suggestions for Authors

very well written and designed study

Point 1: Line 53, 'Meanwhile, progestins are able' not 'is able'

Response 1: Many thanks for your suggestion. We have corrected the error and carefully checked the spelling and grammar of the full text.

Point 2: For the statistics you likely did an ANOVA to detect a difference exists in the group but then did a t-test between each group to detect what that difference is, correct?  Just state that you also did the t-test.  If you are comparing test results to each other use a Tukey correction but if comparing to zero you may wish to use a Dunnetts correction.

Response 2: Thanks so much for your great comments. In this study, we used Graphpad 6.02 to analyse the data.  Tukey’s multiple comparisons test was used as the post hoc test when the differences among multiple groups were compared. Since it is used to test all pairwise comparison among means, we did not do t test and Dunnetts correction after Tukey’s multiple comparisons test. We have added a description in the revised manuscript in lines 591-592.

Reviewer 3 Report

1. I request the authors to add some more literature citing the importance of the work in the area of endometrial cancer

2.  It would be good if the authors provide the percentage of cell population in the flow cytometry figures

3. I would suggest the authors to include clinical importance of this study

4. Overall, the authors provided convincing  data for the biological question they have selected

Author Response

Comments and Suggestions for Authors

Overall, the authors provided convincing data for the biological question they have selected

Point 1: I request the authors to add some more literature citing the importance of the work in the area of endometrial cancer

Response 1: Thank you very much for your helpful comments. In the revised manuscript, we added more literatures about the epidemic situation and development of treatment of endometrial cancer in introduction to emphasize the importance of our study. (Reference 2,3,4 and 9). Please see line 37-39 and 46-50.

Point 2:   It would be good if the authors provide the percentage of cell population in the flow cytometry figures

Response 2: Thank you very much for your constructive advice. Accordingly, we provided the percentage of cell population in the flow cytometry figures (Figure 3A-D and Figure 4C-D).

Point 3:   I would suggest the authors to include clinical importance of this study

Response 3: Thanks for your great comments. We emphasized the clinical importance of this study in lines 394-396, 453-454 and 483-484 in the revised manuscript.